# MaskConvNet: Training Efficient ConvNets from Scratch via Budget-constrained Filter Pruning

## Abstract

In this paper, we propose a framework, called MaskConvNet, for ConvNets filter pruning. MaskConvNet provides elegant support for training budget-aware pruned networks from scratch, by adding a simple mask module to a ConvNet architecture. MaskConvNet enjoys several advantages - (1) **Flexible**, the mask module can be integrated with any ConvNets in a plug-and-play manner. (2) **Simple**, the mask module is implemented by a hard Sigmoid function with small number of trainable mask variables, adding negligible memory and computational overheads to the networks during training. (3) **Effective**, it is able to achieve competitive pruning rate while maintaining comparable accuracy with the baseline ConvNets without pruning, regardless of the datasets and ConvNet architectures used. (4) **Fast**, it is observed that the number of training epochs required by MaskConvNet is close to training a baseline without pruning. (5) **Budget-aware**, with a sparsity budget on target metric (e.g. model size and FLOP), MaskConvNet is able to train in a way that the optimizer can *adaptively* sparsify the network and *automatically* maintain sparsity level, till the pruned network produces good accuracy and fulfill the budget constraint simultaneously. Results on CIFAR-10 and ImageNet with several ConvNet architectures show that MaskConvNet works competitively well compared to previous pruning methods, with budget-constraint well respected. Code is available at https://www.dropbox.com/s/c4zi3n7h1bexl12/maskconv-iclr-code.zip?dl=0. We hope MaskConvNet, as a simple and general pruning framework, can address the gaps in existing literate and advance future studies to push the boundaries of neural network pruning.

## 1 Introduction

State-of-the-art Convolutional Neural Networks (ConvNets) achieve high accuracy in many applications using very deep and wide networks, e.g., VGG Simonyan & Zisserman (2015) and ResNet He et al. (2016). Deep and wide networks are often over-parameterized, which significantly increase the usage of memory (with tens of millions of weights) and compute (with tens of billions of MAC - Multiplication and ACcumulation). It is particularly challenging to deploy, in a naive manner, such large ConvNets on edge device that with limited memory and compute capacities.

To reduce the memory footprint and MAC of modern ConvNets, recent works propose either unstructured pruning (a.k.a. weight pruning) to prune redundant weights LeCun et al. (1990); Han et al. (2015; 2016); Zhu & Gupta (2017); Guo et al. (2016); Louizos et al. (2017a;b); Frankle & Carbin (2019); Lee et al. (2019); Gale et al. (2019) or structured pruning (e.g. filter-wise, channel-wise, depth-wise, etc.) to prune filters/channels/layers Li et al. (2016); Wen et al. (2016); Liu et al. (2017); Ye et al. (2018); Neklyudov et al. (2017); Gordon et al. (2018); He et al. (2018); Véniat & Denoyer (2018); He et al. (2019); Lemaire et al. (2019); Li et al. (2019); Yang et al. (2019) from original over-parameterized deep networks, without incurring significant loss in accuracy. Both unstructured and structured pruning approaches support pruning from pre-trained models followed by fine-tuning and pruning while training from scratch.

Despite the impressive pruning rates achieved on target metrics such as weight variables and floating-point operations (FLOP), existing neural network pruning methods still suffer one or both of the weaknesses that (1) Additional (iterative) fine-tuning is often required to compensate the accuracy degradation, leading to slower convergence and thus more training epochs and (2) Most of existing pruning approaches are not capable of automatically and optimally allocating pruning rates across layers given a sparsity budget on target metric. On the other hand, pruning with budget constraints tailored for specific hardware platforms, as a promising direction for hardware-software co-optimization, is becoming increasingly important especially for edge device.

In this work, we propose a new pruning framework, called MaskConvNet, to address the above mentioned issues simultaneously. MaskConvNet provides elegant support for training budget-aware pruned networks from scratch, by adding a simple mask module to each bundled block of the target ConvNets. Fig. 1 showcases an example of applying MaskConvNet to a typical bundled block Conv/BN/ReLU for filter pruning. The mask module is composed of a hard Sigmoid activation function with a vector of trainable variables as input and the output is regularized with a sparsification loss, resulting in a sparse mask vector with many 0 elements and the rest are real values in $(0, 1]$. The pruned ConvNets is generated by element-wise multiplication between the mask vector and filters/BN parameters, in which filters/BN parameters with all zero are removed from the ConvNets.

We summarize the main contributions and advantages of MaskConvNet as follows.

- **Flexible**. The mask module can be integrated with any ConvNets in a plug-and-play manner.
- **Simple**. Instead of directly regularizing filters, the mask module is defined separately and differentiable almost everywhere, and the number of mask variables equals to the number of filters in the target ConvNets, implying that it adds negligible memory and computational overheads to the networks during training.
- **Effective**. MaskConvNet is able to achieve competitive pruning rate while maintaining comparable accuracy with the baseline ConvNets without pruning, regardless of the datasets and ConvNets architectures used.
- **Fast**. Our empirical observations on both CIFAR10 and ImageNet show that the number of training epochs required by MaskConvNet is close to training a baseline without pruning, mainly attributed to the fact that the mask module (1) enables threshold-free pruning, thanks to the hard Sigmoid activation function and (2) is shared within the bundled block structure (Fig. 1) which in turn addresses the mismatch problem between CONV and BN layers.
- **Budget-aware**. With a sparsity budget on target metric (in percentage), MaskConvNet is able to train in a way that the optimizer can *adaptively* sparsify the network and *automatically* maintain sparsity level, till the pruned network produces good accuracy and fulfill the budget constraint simultaneously. In contrast, previous budget-aware pruning methods suffer the iterative optimization process of alternating between training and evaluation of the target metric, which is tedious and time-consuming.

In recent years, structured pruning attracts much more attention than its unstructured counterpart - weight pruning, mainly because structured pruning doesn't require specific circuit design and thus is more friendly to existing hardware. Thus, in this work we apply MaskConvNet to one of most popular structured pruning methods, filter-wise pruning. It's worth noting that MaskConvNet, as a general pruning framework, can be easily extended to other structured pruning categories as well as weight pruning, by re-formatting the mask variables only. We would like to leave it for future work.

## 2 RELATED WORK

Both unstructured and structured pruning can be grouped into saliency-based LeCun et al. (1990); Han et al. (2015; 2016); Zhu & Gupta (2017); Guo et al. (2016); Frankle & Carbin (2019); Lee et al. (2019); Li et al. (2016); He et al. (2018; 2019) and sparsity learning Louizos et al. (2017a;b); Wen et al. (2016); Liu et al. (2017); Ye et al. (2018); Neklyudov et al. (2017); Gordon et al. (2018); Lemaire et al. (2019); Li et al. (2019) approaches, where the former shrinks network size by well-defined thresholding criterion such as magnitude for weight variables Han et al. (2015; 2016); Zhu & Gupta (2017); Frankle & Carbin (2019); Gale et al. (2019) or L1/L2 norm for filters Li et al. (2016); He et al. (2018; 2019) and the latter prunes networks by either adding regularization functions into training loss (e.g. L0/L1 Louizos et al. (2017a;b); Liu et al. (2017); Ye et al. (2018); Gordon et al. (2018); Lemaire et al. (2019), Group Lasso Wen et al. (2016)) or learning dropout rates for weights/filters

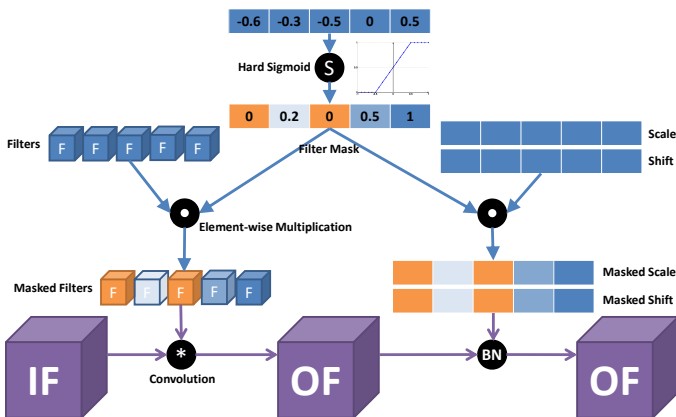

Figure 1: Overview framework of MaskConvNet for filter pruning on a typical bundled block structure composed of Conv - BN - ReLU.

via variational dropout with the reparametrization trick Neklyudov et al. (2017). MaskConvNet falls in the category of sparsity learning with a specially designed regularization function added to the training loss.

Most of the existing pruning approaches, including pruning while training Zhu & Gupta (2017); Li et al. (2016); Liu et al. (2017); Ye et al. (2018); Gordon et al. (2018); He et al. (2018; 2019); Lemaire et al. (2019), require to perform additional (iterative) fine-tuning to compensate the accuracy degradation, mainly due to hard pruning via thresholding and the mismatch between Batch Normalization and its corresponded pruned CONV/FC layer. This leads to slower convergence and thus more training epochs. For instance, with the method of gradual magnitude-based weight pruning during training proposed by Zhu & Gupta (2017) and pre-defined pruning rate 80% on ResNet50, Gale et al. (2019) reported that it increases the number of training epochs by $1.5\times$ in order to achieve comparable ImageNet validation accuracy with the baseline model trained without pruning.

On the other hand, there is few work studying pruning mechanism to automatically and optimally allocate pruning rates across layers that fulfill a budget specified for target metric. Existing works either only meet the budget by pruning using sub-optimal pruning rate pre-defined or empirically calculated Han et al. (2015); Li et al. (2016); Liu et al. (2017) or can derive automatic pruning rate through sparsity learning methods which fail to fulfill the budget explicitly Louizos et al. (2017a;b); Wen et al. (2016); Neklyudov et al. (2017).

MorphNet Gordon et al. (2018) and BAR Lemaire et al. (2019) are the most related work to MaskConvNet, while our work differs from them in several aspects. First, both MorphNet and BAR require pre-defined threshold for hard pruning, while MaskConvNet is threshold-free. Second, MorphNet and BAR apply mask on $\gamma$ variables of BN and filters respectively, while MaskConvNet applies mask on both. Third, to meet the sparsity budget, both MorphNet and BAR alternate between training and evaluation of target metric with a multiplier derived by grid search. MaskConvNet automates the process by adaptively shrinking or expanding the pruning ratio across layers in a data-driven manner during training, avoiding the tedious iterative optimization process. Finally, BAR uses Knowledge Distillation Hinton et al. (2015) to facilitate the training of pruned networks for better accuracy, while MaskConvNet does not.

## 3 METHOD

### 3.1 FILTER PRUNING VIA THE MASK MODULE

Fig. 1 presents an overview of the design of the mask module and how the mask module is applied for filter pruning. The mask module is simply defined as a hard Sigmoid activation function,

$$\tilde{\sigma}(\mathbf{m}) = \min(\max((\mathbf{m}) + 0.5, 0), 1), \tag{1}$$

where input $(\mathbf{m})$ is trainable mask variables tied to Conv filters [1]. For a Conv layer with $w$ input channels, $h$ output channels (filters) and $k \times k$ sized kernels, the number of mask variables equals to $h$. Each element of the mask activation $\tilde{\sigma}(\mathbf{m})$ indicates the importance score of the associated filter. The same mask is applied to bias parameters for the Conv layer.

Each filter is reparameterized as its weights multiplied by its mask activation via element-wise multiplier. Concretely, given a filter $\mathbf{W} \in \mathbb{R}$ and its mask activation $\tilde{\sigma}(\mathbf{m}) \in \mathbb{R}$, the masked filter $\hat{\mathbf{W}}$ is given by:

$$\hat{\mathbf{W}}(\mathbf{W}, \mathbf{m}) = \mathbf{W} \odot \tilde{\sigma}(\mathbf{m}), \qquad (2)$$

where $\odot$ is element-wise multiplication between a mask element and a 3D tesor with size $w \times k \times k$. It's easy to know that mask element with importance score 0 means a filter with $w \times k \times k$ weight parameters is pruned from the network. The sparser the mask activation, the higher the pruning ratio.

It is worth noting that the zero-centered hard-Sigmoid $\tilde{\sigma}(\mathbf{m})$ clamps mask to $[0, 1]$, allowing mask to span the whole range rather than binary. Thus, thresholding critiria is no longer required for hard pruning. The mask module is similar in spirit to the diversity networks Zelda Mariet (2016) in the sense that the pruned filters are "fused" into the remaining filters via non-zero mask elements rather than dropped completely, but our proposal is much simpler.

In our experiments, $\mathbf{m}$ is simply initialized with zeros, so that $\tilde{\sigma}(\mathbf{m}) = 0.5$, which is median of interval $[0, 1]$. One can initialize with random values, but such initialization could potentially give unknown prior bias to sparsification, which may give different structure with different random seed.

In the end of training, the pruned ConvNets is generated by Equation 2, where filters/BN parameters with all zero are removed from the ConvNets. One may note that the output feature maps of Conv layer $i$ serve as the input to Conv layer $i + 1$. Thus, the input channels of layer $i + 1$ that corresponded to the pruned filters of layer $i$ are also pruned.

**Mask Sharing**. In modern ConvNets, a Conv layer is often followed by a batch-norm layer which has "scale" weight parameter $\gamma$ and "shift" bias parameter $\beta$ with dimension length equals to number of filters of its preceding Conv layer.

Previous pruning approaches often prune Conv filters with their successive Batch Normalization layer unchanged. This causes mismatch problem between pruned Conv and BN layers, given that by nature BN parameters highly depend on the output feature maps from preceding Conv. To avoid the mismatch problem, we propose to share the mask module within the bundled block structure Conv-BN-ReLU. More specifically, BN scale and shift parameters are masked with the same mask activation for the corresponding Conv layer. Without mask sharing, we observed that the accuracy degrades severely after pruning. We speculate that this is due to BN by default are unaware of pruned filters, i.e. pruned filters give 0-valued output as input to BN, but BN considers 0-valued input as if the filter is still alive (i.e not pruned).

**Extending Mask to Residual Connections**. In ConvNet architectures such as ResNet, residual connections involve element-wise addition operation where two operands must have the same dimensions. To extend the mask module to residual connections, one can share a single mask for all layers involved in residual connections. This is achieved by taking the mask for the last Conv layer in the last residual block as the base mask, and sharing the base mask with other Conv layers involved in other residual connections. Partially, the base mask is sliced to fit Conv layers with less number of filters.

### 3.2 SPARSITY REGULARIZATION ON THE MASK

With filters $\mathbf{W}$ reparameterized by the mask activation $\tilde{\sigma}(\mathbf{m})$, the loss function for training budget-aware pruned ConvNets from scratch is given as:

$$\mathcal{L}(\mathbf{W}, \mathbf{m}) = \mathcal{L}_E(\mathbf{W} \odot \tilde{\sigma}(\mathbf{m})) + \mathcal{L}_S(\tilde{\sigma}(\mathbf{m})), \qquad (3)$$

where $\mathcal{L}_E$ is cross-entropy loss for classification (the "error"), and $\mathcal{L}_S$ is the sparsification loss defined on the mask.

---

[1]We take Conv/FC as example here as they account for over 99% of total operations in modern ConvNets. FC can be treated as a special case of Conv layer where the kernal size is $1 \times 1$.

**Sparsification Loss**. The sparsification loss is defined by sparsifying the mask:

$$\mathcal{L}_S(\tilde{\sigma}(\mathbf{m})) = \lambda_m \mu(\tilde{\sigma}(\mathbf{m})) - \lambda_v \frac{\sigma^2(\tilde{\sigma}(\mathbf{m}))}{\mu(\tilde{\sigma}(\mathbf{m}))}, \tag{4}$$

where the goal is to minimize the mean $\mu(\tilde{\sigma}(\mathbf{m}))$, and maximize variance-to-mean $\sigma^2(\tilde{\sigma}(\mathbf{m}))/\mu(\tilde{\sigma}(\mathbf{m}))$. In particular, the regularization multipliers $\lambda_m$ and $\lambda_v$ are introduced to precisely control regularization strength towards budget-aware pruning (see details in the subsequent subsection).

Minimizing mean is basically L1-regularization averaged over the total number of mask elements of the network, and maximizing variance is soft-binarization by dispersing mask elements to the either zero or one. Dividing variance by mean adaptively strengthen variance maximization when mean is becoming smaller.

**Extension to Multi-metrics**. Equation 3 can be extended to support sparsity regularization of the mask for multiple target metrics simultaneously, e.g. optimizing for parameter counts and FLOPs given by budget constraints for both metrics. A simple heuristic is to average the mean and variance-to-mean of the mask $\tilde{\sigma}(\mathbf{m})$ that is applied to $N$ target metrics. Accordingly, Equation 3 and Equation 4 are rewritten as $\mathcal{L}(\mathbf{W}, \mathbf{m}) = \mathcal{L}_E(\mathbf{W} \odot \tilde{\sigma}(\mathbf{m})) + \sum_{i=1}^{N} \mathcal{L}_{Si}(\tilde{\sigma}(\mathbf{m}))$ and $\sum_{i=1}^{N} \mathcal{L}_{Si}(\tilde{\sigma}(\mathbf{m})) = \sum_{i=1}^{N} \lambda_{mi} \cdot \mu(\tilde{\sigma}(\mathbf{m})) - \sum_{i=1}^{N} \lambda_{vi} \cdot \frac{\sigma^2(\tilde{\sigma}(\mathbf{m}))}{\mu(\tilde{\sigma}(\mathbf{m}))}$, respectively.

---

**Algorithm 1:** Algorithm description of MaskConvNet for 1 epoch.

---

**Input:** input-label pairs $(\mathbf{x_i}, \mathbf{y_i})$ from $N$ training mini-batches
**Output:** Weights $\mathbf{W}$, mask $\mathbf{m}$
Set target sparsity percentage $s^*$, base multipliers $\lambda_m^{base}$ and $\lambda_v^{base}$, multiplier update interval $n$,
 multiplier EMA exponent $\alpha$;
**for** $i$ in 1, 2, ... N **do**
 $(\mathbf{x_i}, \mathbf{y_i}) \leftarrow$ load_mini_batch($i$);
 // For every $n$ mini-batches, evaluate sparsity and update multipliers;
 **if** ($i \% n == 0$) **then**
  $s_t \leftarrow$ compute_sparsity($\mathbf{W} \odot \tilde{\sigma}(\mathbf{m})$), according to Eqn. 2 ;
  $s_t \leftarrow s_t^{\alpha} + s_{t-1}^{(1-\alpha)}$ ;
  $\Delta s \leftarrow s^* - s_t$;
  $\lambda_m \leftarrow \lambda_m^{base} \times \Delta s$;
  $\lambda_v \leftarrow \lambda_v^{base} \times \Delta s$;
 **end**
 Train on $(\mathbf{x_i}, \mathbf{y_i})$ with cross-entropy and mask regularization as in Eqn. 3;
**end**

---

## 3.3 BUDGET-AWARE MASKCONVNET: THE ALGORITHM

In previous works on sparsity-regularization based pruning Gordon et al. (2018); Lemaire et al. (2019), grid search on sparsity regularization multiplier is performed, the multiplier is then used to find a sparse network structure that could meet the budget. These approaches alternate between network training and sparsifying with the regularization multiplier until sparsification stops at arbitrary sparsity level. If the model is undersparsed or oversparsed, the regularization multiplier value is adjusted and training is repeated from scratch. This is very laborious and time-consuming.

In contrast to previous works, MaskConvNet avoids the tedious iterative optimization process. With a target sparsity budget (in percentage), MaskConvNet is able to train in a way that the optimizer *adaptively* sparsifies the network and *automatically* maintains sparsity level, till the pruned network produces good accuracy and fulfills the budget constraint simultaneously.

The training procedure for one epoch is described in Algorithm 1. During training, the weight tensors are reparameterized with masks according to Eqn. 2, both weight tensors and masks are jointly updated and sparsified via standard optimizer (e.g. SGD with momentum) according to Eqn. 3. Evaluating sparsity percentage at every mini-batch iteration can slow down training considerably,

hence we evaluate and smooth the sparsity (with EMA) for every $n$ interval. In our experiments, we set $n = 20$ and EMA exponent $\alpha = 0.99$.

As the regularization multipliers are periodically updated, sparsification is expanded stronger in early training, and becomes gradually weaker when reaching the target due to $\Delta s$ approaches zero. On the other hand, when the pruned model is oversparsed, $\Delta s$ becomes negative, and so do the multipliers $\lambda_m$ and $\lambda_v$. The optimizer will then shrink the mask sparsity (unsparsification) until $\Delta s$ is positive again (See more detailed analysis in the next subsection).

### 3.4  How Mask Unsparsification Works?

In case when $\lambda_m$ and $\lambda_v$ are too big, too many mask elements will be zeroed and the model becomes *oversparsified* and misses the sparsity budget. If the masks elements are zeroed, how would the optimizer continue update these zeroed elements? How would the gradients flow and update the masks? We exploit three things to unsparse the masks: (1) Mask decay update; (2) Negative regularization multiplier; and (3) Positive mask parameter when hard-Sigmoid output is zero.

The SGD update rule for the regularization term $\mathcal{L}_S(m)$ is given as $m \leftarrow m - \lambda_m \eta \frac{\partial S}{\partial m} - \lambda_m \eta \epsilon m$, where $\eta$ is the learning rate and $\epsilon$ is mask decay constant. Also recall the hard-Sigmoid function on mask is $\tilde{\sigma}(m) = \min(\max(m + 0.5, 0), 1)$.

$\lambda_m$ will be negative when sparsity difference $\Delta s$ is negative. Also, when $m = -0.5$, then $\tilde{\sigma}(m) = 0$ and $\frac{\partial S}{\partial m} = 0$. However, $m$ is still non-zero negative. Hence, the term $-\lambda_m \eta \epsilon m$ is positive , and thus the optimizer is able to increase $m$ so that $\tilde{\sigma}(m)$ becomes non-zero, unsparsifying the mask.

The catch here is that we cannot achieve exactly zero error to target sparsity budget due to zeroed $\tilde{\sigma}(m)$ will always be bounced back to non-zero (unless we deliberately turn off mask decay). The model will always be undersparsfied and oversparsified continuously by small degree throughout training. The sparsity deviation depends on the value of mask decay constant applied to the mask. This will be discussed in experimental section.

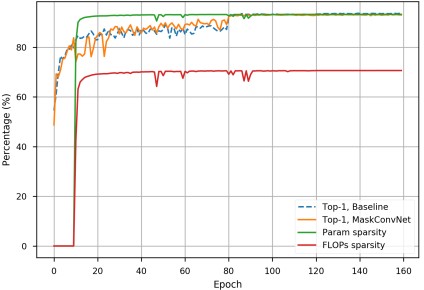 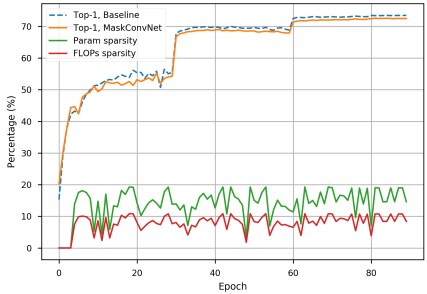

Figure 2: Testing/Validation accuracy (Top-1) and sparsity pattern as the function of training epochs for MaskConvNet with WideResNet-28-10 on CIFAR-10 (left) and ResNet-34 on ImageNet (right). For reference, the baseline accuracy is included.

## 4  Experiments

We evaluate MaskConvNet on both CIFAR-10 and ImageNet ILSVRC 2012 et al. (2009), with different ConvNets architectures including Simonyan & Zisserman (2015); He et al. (2016); Zagoruyko & Komodakis (2016) as baseline architectures. For all CIFAR-10 experiments, we modified source code from Liu et al. (2018), and followed the same training setup and data augmentation. We used SGD with momentum of $0.9$ as optimizer. Initial learning rate is set to $0.1$ and divided by 10 at epoch 40 and epoch 80. All models are trained for 160 epochs. $\lambda_m$ and $\lambda_v$ are updated every 20 mini-batches with EMA exponent $\alpha = 0.99$. Details on the settings of $\lambda_m$ and $\lambda_v$ are left in appendix section. For all ImageNet experiments, we implemented the training code and hyper-parameter settings derived from source code for work by Banner et al. (2018).

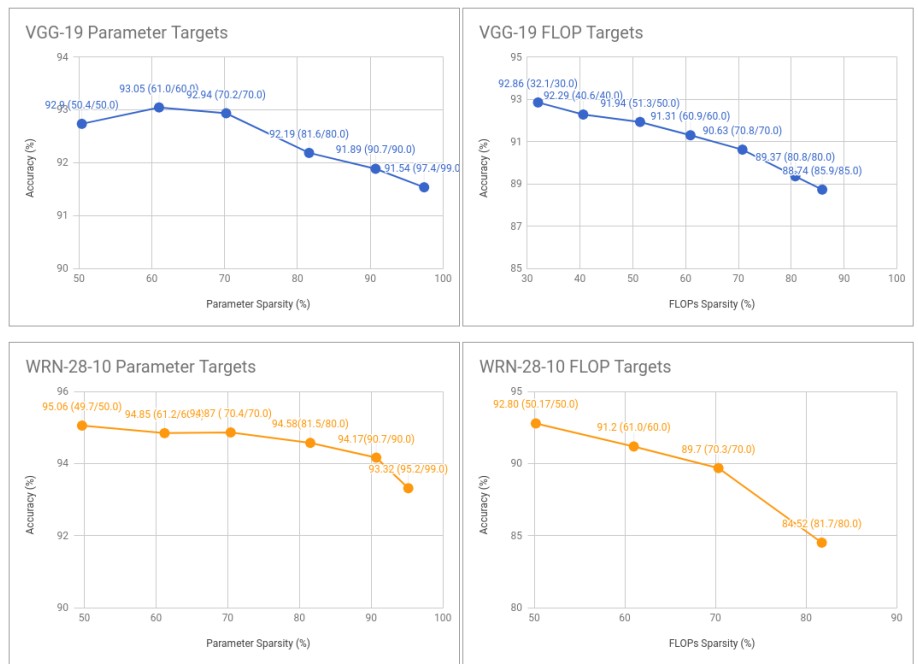

Figure 3: Results of MaskConvNet-P (parameter counts as target metric) and MaskConvNet-F (FLOP as target metric) with various sparsity budgets on CIFAR-10, using VGG-19 and WideResNet-28-10 as baseline architectures. Text labels on plot points denote "Accuracy% (Actual / Budget Sparsity%)

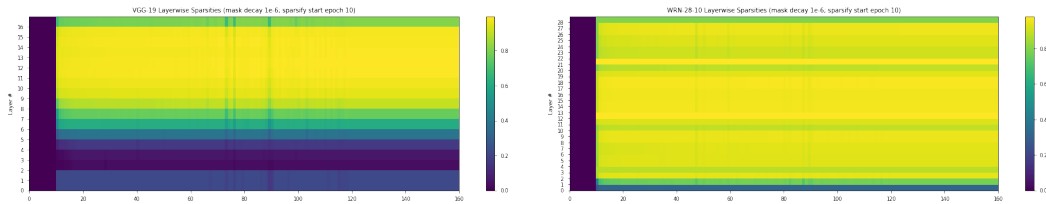

Figure 4: Layer-wise parameter sparsities throughout training for VGG-19 (left) and WideResnet-28-10 (right) on CIFAR-10, with warmup training till epoch 10. The brigher the color, the higher the sparsity.

## 4.1 MASKCONVNET: IN-DEPTH ANALYSIS

In this section, we perform various experiments on CIFAR-10 and ImageNet to explore the behaviors of MaskConvNet in-depth.

**The convergence speed and sparsity pattern of MaskConvNet during training**. Figure 2 shows the testing/validation accuracy (Top-1) and sparsity pattern as the function of training epochs for MaskConvNet with WideResNet-28-10 on CIFAR-10 and ResNet-34 on ImageNet. The baseline accuracy is included for reference. We observe that the convergence speed of MaskConvNet is highly similar to the baseline architecture, indicating the effective training capability of MaskConvNet. In addition, the sparsity pattern emerges at early training epochs and becomes stable as training going on, which is consistent with the observation from the single-shot pruning work Lee et al. (2019) that redundant neurons can be identified in a single step even before training. One may note that the sparsity curve for ResNet-34 has small fluctuations. As shown in the next subsection as well as appendix, this issue can be solved by proper choice of mask decay.

**Can MaskConvNet meet the Budget?**. Figure 3 presents the results of MaskConvNet-P (parameter counts as target metric) and MaskConvNet-F (FLOP as target metric) with various sparsity budgets

on CIFAR-10, using VGG-19 and WideResNet-28-10 as baseline architectures. Accuracy almost linearly decreases with increasing sparsity for both target metrics. More importantly, it is worth noting that the actual sparsities produced by both MaskConvNet-P and MaskConvNet-F are close to the budget, i.e. the error between actual and budget sparsity is within 1% (absolute value) for most of the operating points.

**Looking into layer-wise parameter sparsities**. We also investigate layer-wise parameter sparsity patterns throughout training on CIFAR-10, for VGG-19 trained with 50% FLOP sparsity budget and WideResnet-28-10 trained with 93.75% parameter sparsity budget. Figure 4 shows VGG-19 and WideResnet-28-10 training results respectively. We observe that early layers are less likely to be sparsified, which matches the intuition in previous works such as Gordon et al. (2018). The hyper-thesis is that early layers carry more information and can incur bigger accuracy loss if sparsified too much. In addition, we noted more uniform overall sparsity on WideResNet-28-10 than VGG-19. We speculate this to be related to role of residual connections during sparsification process. This phenomenon is left for future investigation.

More results and analysis are introduced in the appendix section, including the sensititity analysis of base regularization multipliers $\lambda_m$ and $\lambda_v$, visualization of mask histogram output by the hard-Sigmoid, inference speedup by measuring wall-clock latency of the baseline and pruned models on a CPU core, and the effect of mask decay and warmup on training accuracy as well as layer-wise sparsity pattern.

Table 1: Comparison of MaskConvNet-F or MaskConvNet-P with state-of-the-art pruning approaches Network Slimming Liu et al. (2017), L1-Pruning Li et al. (2016) and BAR Lemaire et al. (2019), on CIFAR-10 with various ConvNet architectures.

| Model | Accuracy (%) | FLOPs | Sparsity (%) | Parameters (M) | Sparsity (%) |
|---|---|---|---|---|---|
| **VGG-16** | | | | | |
| Baseline | 93.63 | $3.15 \times 10^8$ | 0.00 | 14.99 | 0.00 |
| L1-Pruning | 93.40 | $2.06 \times 10^8$ | 34.20 | 5.40 | 64.00 |
| MaskConvNet-F | 93.40 | $1.88 \times 10^8$ | **40.19** | 1.72 | **88.53** |
| **VGG-19** | | | | | |
| Baseline | 93.67 | $7.97 \times 10^8$ | 0.00 | 20.04 | 0.00 |
| Network Slimming | 93.80 | $3.91 \times 10^8$ | 51.00 | 2.30 | 88.5 |
| MaskConvNet-F | 93.03 | $3.66 \times 10^8$ | **54.08** | 1.51 | **92.47** |
| **ResNet-56** | | | | | |
| Baseline | 93.40 | $2.52 \times 10^8$ | 0.00 | 0.85 | 0.00 |
| L1-Pruning | 93.06 | $9.09 \times 10^7$ | **27.60** | 0.73 | 13.7 |
| MaskConvNet-F | 92.25 | $1.88 \times 10^8$ | 25.27 | 0.58 | **31.87** |
| **ResNet-164** | | | | | |
| Baseline | 94.70 | $5.02 \times 10^8$ | 0.00 | 1.71 | 0.00 |
| Network Slimming | 94.92 | $3.81 \times 10^8$ | 23.70 | 1.44 | 14.9 |
| MaskConvNet-F | 94.28 | $3.46 \times 10^8$ | **30.96** | 0.79 | **53.59** |
| **WideResNet-28-10** | | | | | |
| Baseline | 95.30 | $1.19 \times 10^{10}$ | 0.00 | 36.48 | 0.00 |
| BAR ($16\times$ V reduction) | 91.62 | $1.74 \times 10^8$ | **98.54** | 2.31 | **93.67** |
| MaskConvNet-P | 94.28 | $3.50 \times 10^9$ | 70.63 | 2.46 | 93.25 |

Table 2: Comparison of MaskConvNet-FP with both parameter counts and FLOP as target metrics with Li et al. (2016) on ImageNet ILSVRC 2012 with ResNet-34 as baseline architecture.

| Model | Accuracy (%) | FLOPs | Sparsity (%) | Parameters (M) | Sparsity(%) |
|---|---|---|---|---|---|
| **Resnet-34** | | | | | |
| Baseline | 73.27 | $3.69 \times 10^9$ | 0.00 | 21.8 | 0.00 |
| L1-Pruning | 72.56 | $\mathbf{3.08 \times 10^9}$ | **15.50** | 19.9 | 7.60 |
| MaskConvNet-PF | 72.56 | $3.29 \times 10^9$ | 10.75 | **17.7** | **19.00** |

### 4.2 COMPARISON WITH PREVIOUS WORKS

**CIFAR-10**. Table 1 compares MaskConvNet-F or MaskConvNet-P with state-of-the-art pruning approaches Network Slimming Liu et al. (2017), Pruning Filters for Efficient ConvNets Li et al. (2016) denoted as L1-Pruning and Budget-Aware Regularization (BAR) Lemaire et al. (2019), on CIFAR-10 with various ConvNet architectures including VGG-16, VGG-19, ResNet-56, ResNet-164 and WideResNet-28-10. One can see that for almost all of the operating points, MaskConvNet-F performs comparably in terms of accuracy with Network Slimming and L1-Pruning, and achieves much higher sparsity on both FLOP and parameter counts. Compared to BAR, MaskConvNet-P achieves comparable parameter sparsity with better accuracy, but the FLOP sparsity is much lower probably because the usage of Knowledge Distillation Hinton et al. (2015) in BAR as prior guiding the training process while our method does not use it.

**ImageNet**. Table 2 compares MaskConvNet-FP with both parameter counts and FLOP as target metrics with L1-Pruning Li et al. (2016), with ResNet-34 as the baseline architecture. In L1-Pruning paper, Resnet-34 is pre-trained, pruned, and fine-tuned for 20 epochs. Standard ResNet training takes 90 epochs. Compared to L1-Pruning, we managed to train ResNet-34 with fewer epochs ($<90$) that achieves the same accuracy and much higher parameter sparsity, but with slightly lower FLOP sparsity.

## 5 CONCLUSION AND FUTURE WORK

In this work, we proposed MaskConvNet, a general pruning framework where a model is trained and sparsified end-to-end with sparsity budget constraint on various target metrics. Promising results are reported on CIFAR-10 and ImageNet for the problem of filter pruning. We hope MaskConvNet can inspire future research including extension of the pruning framework (1) to other structured pruning categories as well as unstructured pruning and (2) to other model structures beyond ConvNets, such as recurrent networks and self-attention networks.

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

# A APPENDIX

## A.1 HYPER-PARAMETERS FOR SINGLE-METRIC AND MULTI-METRIC EXPERIMENTS (TABLE 3, 4)

Table 3: Hyper-parameters for Single-metric experiments

| Model | VGG-19 | | WideResnet-28-10 | |
|---|---|---|---|---|
| | Target P | Target F | Target P | Target F |
| $\lambda_m^{base}$ | 1 | 5 | 3, 5 $\lambda_v^{base}$ | 5 |
| 6 | 1, 2* | 5 | | |

Table 4: Hyper-parameters for Multi-metric experiments

| | MaskConvNet-P ($\lambda_m^{base}$, $\lambda_v^{base}$, warmup) | MaskConvNet-F ($\lambda_m^{base}$, $\lambda_v^{base}$, warmup) | MaskConvNet-PF ($\lambda_{m1}^{base}$, $\lambda_{v1}^{base}$, $\lambda_{m2}^{base}$, $\lambda_{m2}^{base}$, warmup) |
|---|---|---|---|
| VGG-16 | 3, 4, 20 | 5, 6, 20 | 3, 4, 5, 6, 20 |
| VGG-19 | 3, 4, 20 | 3, 4, 20 | 3, 4, 3, 4, 20 |
| Resnet-56 | 2, 3, 50 | 1, 2, 50 | 2, 3, 1, 2, 50 |
| Resnet-164 | 1, 5, 10 | 1, 5, 10 | 1, 5, 1, 5, 10 |
| Resnet-34 | – | – | 5, 20, 5, 20, 3 |

## A.2 MASK HISTOGRAM

As shown in Figure 5, the mask values accumulate mostly at 0, then at 1. This observation is different compared to findings by Gordon et al. (2018), where the batch-norm scale parameter histogram resembles Gaussian distribution.

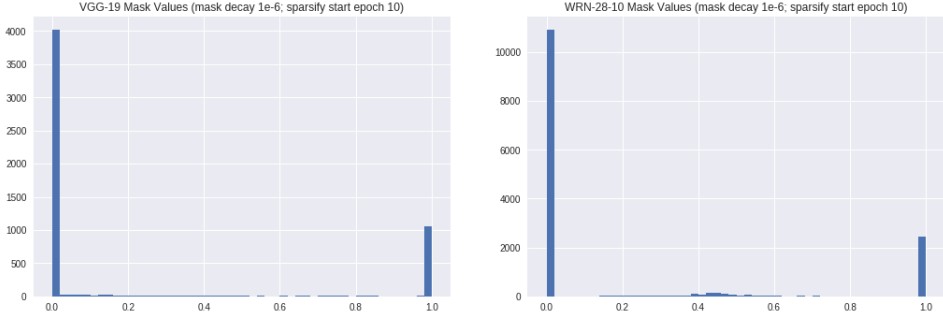

Figure 5: Mask Histogram after training.

## A.3 SENSITIVITY ANALYSIS FOR BASE REGULARIZATION MULTIPLIERS

We train VGG-19 with target FLOP sparsity budget of $50\%$, with base regularization multipliers $\lambda_m^{base}$ and $\lambda_v^{base}$ sampled from integers in interval $[0, 10]$. We perform 10 trials. All models are trained for 160 epochs, and used late sparsity regularization (i.e. warmup) start at epoch 10. Results are shown in Table 5, one can see that the accuracy is quite stable regardless of the base regularization multipliers used.tt

## A.4 LATENCY REDUCTION

We measure wall-clock latency for forward pass of one mini-batch on Intel Xeon E5-2680 V4 Processor (Broadwell-EP, 14 core, 2.4GHz) for CIFAR-10 models. We select checkpoint files for three models with different target FLOP sparsity budgets. Measurements are noisy, but overall we found latencies are reduced compared to the baseline models (Table 6).

Table 5: Sensitivity analysis for base regularization multipliers $\lambda_m^{base}$ and $\lambda_v^{base}$ with VGG-19 trained on CIFAR-10 with target FLOP sparsity 50%

| $\lambda_m^{base}$ | $\lambda_v^{base}$ | Accuracy (%) | FLOPs sparsity (%) | Parameter sparsity (%) |
|---|---|---|---|---|
| 1 | 3 | 92.43 | 49.10 | 91.12 |
| 3 | 9 | 92.10 | 53.48 | 92.93 |
| 7 | 4 | 91.96 | 50.13 | 91.28 |
| 2 | 2 | 92.02 | 46.93 | 89.22 |
| 5 | 9 | 91.75 | 50.03 | 91.36 |
| 4 | 6 | 92.01 | 52.30 | 92.50 |
| 1 | 10 | 92.33 | 44.72 | 87.56 |
| 5 | 7 | 92.50 | 52.66 | 92.77 |
| 9 | 10 | 92.15 | 49.63 | 90.90 |
| 10 | 2 | 92.43 | 51.73 | 92.15 |
| mean $\pm$ std | | $92.17 \pm 0.25$ | $50.07 \pm 2.70$ | $91.18 \pm 1.68$ |

Table 6: Latency Measurements on CIFAR-10

| Model | Target FLOP sparsity (%) | Actual FLOP sparsity (%) | Latency (ms) |
|---|---|---|---|
| VGG-19 | 0 (Baseline) | 0.00 | $22.45 \pm 1.17$ |
| | 50 | $51.91 \pm 2.46$ | $12.94 \pm 0.42$ |
| | 60 | $62.35 \pm 2.92$ | $11.42 \pm 0.28$ |
| | 70 | $69.52 \pm 0.01$ | $10.81 \pm 0.06$ |
| WideResnet-28-10 | 0 (Baseline) | 0.00 | $64.15 \pm 0.21$ |
| | 50 | $54.59 \pm 1.5$ | $25.43 \pm 1.09$ |
| | 60 | $58.54 \pm 2.5$ | $22.79 \pm 0.94$ |
| | 70 | $72.58 \pm 4.2$ | $19.92 \pm 0.93$ |

## A.5 EFFECT OF MASK DECAY AND WARMUP

Figure 6 shows different mask decays give different rate of unsparsification. Also with warmup training (late sparsification), accuracy is improved and the pruned structure becomes different compared to the one without warmup - High contrast is visible between sparsity level between last layers and early layers. Last layers are sparsified more.

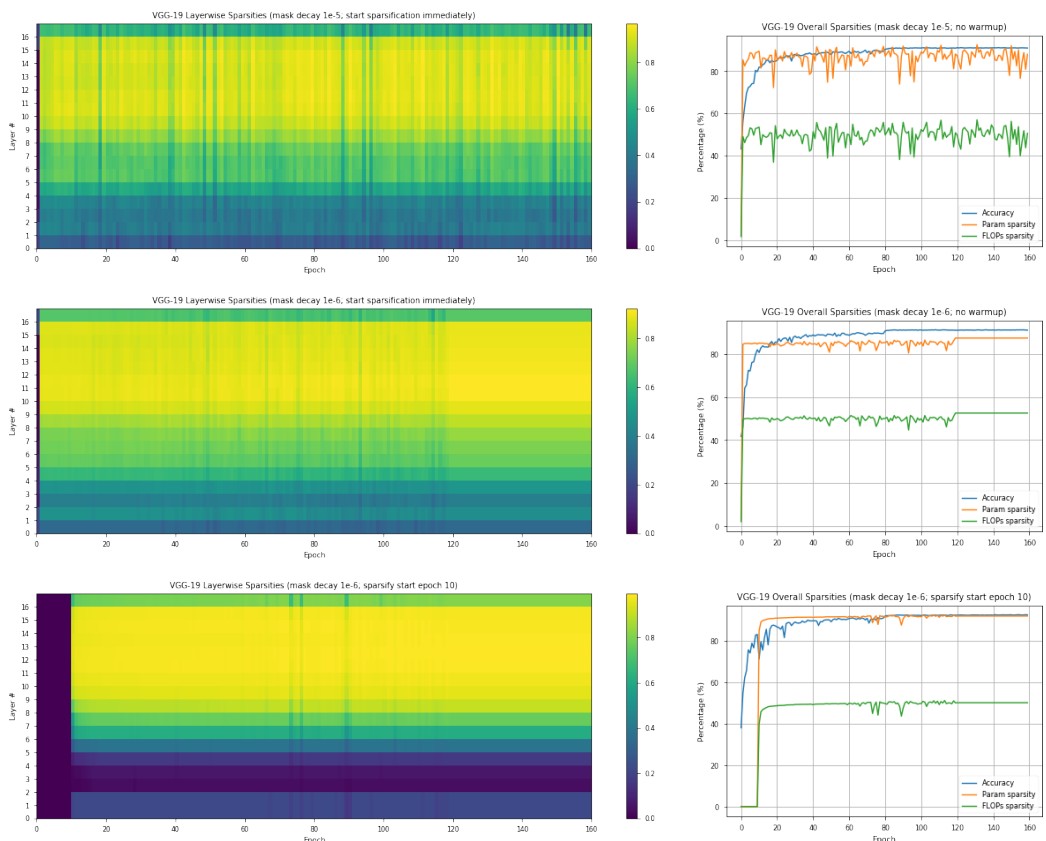

Figure 6: Layer-wise parameter sparsities throughout training for VGG-19 on CIFAR-10, with different mask decay and warmup training. The brigher the color, the higher the sparsity.

