# OpenReview forum: "MaskConvNet: Training Efficient ConvNets from Scratch via Budget-constrained Filter Pruning"
_ICLR.cc/2020/Conference — Reject_

### Official Review · AnonReviewer3 · 2019-10-23
**Official Blind Review #3**

**Rating:** 3

**Review:**

This paper proposes a framework for training time filter pruning for convolutional neural networks. The main idea is to use a trainable soft binary mask to zero out convolutional filters and corresponding batch norm parameters.

Pros:
+ The proposed method seems relatively easy to implement.
+ The quantitative results in the paper indicate that MaskConvNet achieves performance competitive with previously proposed pruning methods.

Cons:
- Writing of the paper could be significantly improved. See some examples below.
- The main thing that bothered me about the method was the usage of hard sigmoid. If a mask component ever gets into one of the flat regions it won’t be able to escape. The authors propose a workaround which they call “mask decay update”. This approach looks quite hacky and I’m not sure how easy it is to make it work in practice.

Notes/questions:
* Abstract: “elegant support” -> “support”
* Everywhere in the text: Back-to-back citations should have the form (citation1; citation2; …)
* Section 1, paragraph 3: “suffer one” -> “suffer from one”
* Section 1, paragraph 4: “above mentioned” -> “above-mentioned”
* Figure 1: The figure would greatly benefit from a detailed description. What’s IF, OF and OF? The reader shouldn’t be guessing.
* Section 2, paragraph 3: “corresponded” -> “corresponding”
* Section 3.1, paragraph 2: “W \in R” – W is probably not a scalar value therefore it’s in R^n. The same goes for the mask.
* Section 3.1, paragraph 2: “It’s easy to know ...” – this sentence needs to be rewritten, e.g., “One can see that …”
* Section 3.1, paragraph 2: “sparser” -> “more sparse”
* Section 3.2, “Extension to Multi-metrics”: “FLOPs” are never defined in the paper. How is this quantity computed exactly? I’m also not entirely sure how useful it is to introduce multiple lambdas – it seems that this case corresponds to a single lambda which is a sum \lambda_i.
* Section 3.3, paragraph 1: “undersparsed”, “oversparsed” – not sure if these words exist. Maybe rephrase instead of introducing new terms?
* Section 3.3, paragraph 1: “very laborious” -> “laborious”
* Figure 3: Why not show points all the way to 0 sparsity?
* Section 4.2, CIFAR-10: The authors mention that (Lemaire et al., 19) achieve better FLOP sparsity due to usage of Knowledge Distillation. From this sentence alone it’s not clear how exactly KD helps. Why can’t KD be applied in the proposed framework? I’d appreciate if the authors could elaborate on this.

I must admit that I’m not an expert in the field of NN pruning but I’m surprised that training-time masking of filters has not been tried before. Even if it’s really the case I’m not entirely confident that the paper should be accepted: the “unsparsification” looks more like a hack than a principled approach and the overall quality of writing needs to be improved. I’m giving a borderline score for now but I’m willing to increase it provided that the rebuttal addresses my concerns.

**Experience Assessment:**

I do not know much about this area.

**Review Assessment: Checking Correctness Of Derivations And Theory:**

I assessed the sensibility of the derivations and theory.

**Review Assessment: Checking Correctness Of Experiments:**

I did not assess the experiments.

**Review Assessment: Thoroughness In Paper Reading:**

I read the paper at least twice and used my best judgement in assessing the paper.

---

> ### Author Response · Authors · 2019-11-13
> **Author response to Reviewer #3**
>
>
> Thank you for reviewing our work. We try to correct all the grammatical mistakes and typos. Below we address your main concern.
>
> Q1: The main thing that bothered me about the method was the usage of Hard Sigmoid. If a mask component ever gets into one of the flat regions it won’t be able to escape. The authors propose a workaround which they call “mask decay update”. This approach looks quite hacky and I’m not sure how easy it is to make it work in practice.
>
> Unlike heuristic pruning approaches such as magnitude-based pruning via thresholding, trainable pruning involves differentiating through 0 values during back-propagation, which is a long-standing difficult problem in the literature of network pruning.
>
> If learning mask as discrete binary values {0,1},  one cannot get true gradients during back-propagation, but only estimates under Expectation assumptions. For example, the principled approaches like Variational Dropout [1] and L0-regularization [2] proposed continuous relaxation/re-parameterization to learn the activation probability of the mask. However, this method has "sparse" gradients issue as noted by L0-regularization. BAR [3] used the same re-parametrization trick to learn mask and reported difficulty in getting good accuracy with this technique unless using some tuned linearly decreasing regularization strength and knowledge distillation. For SSL [4], MorphNet [5] and Network Slimming [6], once a weight parameter is set to 0 during training, it is always 0 and can never be re-activated again.
>
> Other options include (1) using REINFORCE to learn the mask values, but this has high variance issue, or (2) using the Straight-Through Estimator to backprop through 0 (i.e., defining the backward pass “gradient” as other functions such as identity function), but this has biased gradients estimate issue, as referred to discussions in [2].
> Instead of learning mask with discrete values, we proposed the soft-binarization approach – MaskConvNet, for differentiating through mostly continuous values on the Hard Sigmoid function, where the mask values are in the interval [0,1] instead of discrete {0,1}.
>
> Our method does not rely on costly and noisy sampling procedures, and not requiring regularization strength scheduling as in [3]. With soft-binarization, the model learns mask outputs with mostly continuous values. When the model is reaching the budget sparsity, the final pruned structure is also found. More importantly, the reverse mask weight-decay is a method to recover dead neurons if the pruning rate exceeds the pre-defined budget. Reverse weight-decay requires non-zero parameters to work, and this is the reason why we use hard-sigmoid: the actual mask parameters stay non-zero while the mask outputs are zero (when mask parameters are smaller than -0.5). In the appendix section, we show that this method is robust to hyper-parameter selections. Mask decay value is set to 1e-5 or less and EMA alpha=0.99 regardless of the network architectures and the datasets used.
>
> We agree that other methods have their appeal due to their theoretical grounding (less "hacky"). However, we can see lots of "hacks" in regularization literature such as Dropout, Label Smoothing, Confidence Penalty, Cutout, Mixup, Knowledge Distillation, BatchNorm, etc. Some took years to be improved and be theoretically sound, e.g. Dropout -> Variational Dropout, MC Dropout, etc. And some still have no 100% agreed consensus or analytically proved on why it works, e.g. Knowledge Distillation and BatchNorm. Likewise, we leave the investigation for the theoretical analysis of reverse weight-decay for future work.
>
> [1] Variational Dropout and the Local Reparameterization Trick https://arxiv.org/pdf/1506.02557.pdf
> [2] Learning Sparse Neural Networks through L0 Regularization  https://arxiv.org/abs/1712.01312
> [3] Structured Pruning of Neural Networks With Budget-Aware Regularization http://openaccess.thecvf.com/content_CVPR_2019/html/Lemaire_Structured_Pruning_of_Neural_Networks_With_Budget-Aware_Regularization_CVPR_2019_paper.html
> [4] Learning Structured Sparsity in Deep Neural Networks https://arxiv.org/abs/1608.03665
> [5] Learning Efficient Convolutional Networks through Network Slimming https://arxiv.org/abs/1708.06519
> [6] MorphNet: Fast & Simple Resource-Constrained Structure Learning of Deep Networks https://arxiv.org/abs/1711.06798

---

> > ### Comment · AnonReviewer3 · 2019-11-15
> > **Comment**
> >
> > Thank you for your response!
> >
> > Minor remark on the paper: When you introduce unsparsification technique you use "S" which you never define in the text. I'm guessing you're referring to "\mathcal{L}_S".
> >
> > In my notes I asked a couple more questions and it would be nice if you could comment on those too.
> >
> > I still don't quite understand how you use use target metrics in Algorithm 1. What about multi-metrics? Do you have any experiments employing this concept? This bit is not very clear. After re-reading the section, I have a guess that you propose to update different lambdas independently. Still, I think the paper would benefit from making this point more explicit.
> >
> > In general, it'd be much easier for me to make a decision based on the content of the revised paper.

---

### Official Review · AnonReviewer2 · 2019-10-23
**Official Blind Review #2**

**Rating:** 3

**Review:**

In this work, the authors propose a network pruning method to learn a pruned network during training. Specifically, they add a pruning mask for each layer and induce a sparisity loss on the mask variables during training. The pruned network is obtained by applying the learned mask to the networks.

The paper seems to be well contained. However, my assessment of this paper is weak reject. I am mainly concerned with the novelty of this method. Also i think some more evaluation is needed to fully understand the effectiveness of this method. My questions are summarized as follows:

Q1: In the methods part, the authors said that “Previous pruning approaches often prune Conv filters with their successive Batch Normalization layer unchanged.” Can the authors give some reference here as to which pruning approaches?

Q2: Did the authors compare the proposed approach to training the pruned networks from scratch as done in [1]? Also can the authors analyze the sparsity patterns of the pruned networks as done in section G in the appendix of [1]?

Q3: What is the difference of your approach to [2]? They seem to be very similar. I think it is necessary to add some discussion in the related work. Is there any experimental results for comparison with [2]?

[1] Rethinking the Value of Network Pruning. Liu et al. ICLR 2019
[2] AutoPruner: An End-to-End Trainable Filter Pruning Method for Efficient Deep Model Inference. Luo et al. Arxiv, 2018.

**Experience Assessment:**

I have published one or two papers in this area.

**Review Assessment: Checking Correctness Of Derivations And Theory:**

I carefully checked the derivations and theory.

**Review Assessment: Checking Correctness Of Experiments:**

I carefully checked the experiments.

**Review Assessment: Thoroughness In Paper Reading:**

I read the paper at least twice and used my best judgement in assessing the paper.

---

> ### Author Response · Authors · 2019-11-13
> **Author response to Reviewer #2**
>
> Thank you for reviewing our work.
>
> Q1: In the methods part, the authors said that “Previous pruning approaches often prune Conv filters with their successive Batch Normalization layer unchanged.” Can the authors give some reference here as to which pruning approaches?
>
> Most of the train-prune-retrain pruning approaches suffer this issue. For example, Network Slimming [3] and MorphNet [4] proposed to sparsify BN scale/shift parameters. However, even though the scale is exactly 0, when they pruned the corresponding filter (i.e. cut down output depth for current Conv layer and input depth for the next Conv layer), the accuracy of the pruned model drops and must be fine-tuned. According to the appendix section of MorphNet, the BN statistics are disrupted after pruning and must be corrected with several thousand iterations with a tiny learning rate.
> Our method masked the filters of Covn/FC layer and the corresponding BN layer with the same mask vector to condition BN to ignore the masked filters during training, thus avoid the requirement for finetuning.
>
>
>
> Q2: Did the authors compare the proposed approach to training the pruned networks from scratch as done in [1]? Also can the authors analyze the sparsity patterns of the pruned networks as done in section G in the appendix of [1]?
>
> We added experiments to compare our method with the train-from-scratch method named Soft Filter Pruning (SFP) [5] with ResNet56 v1 on CIFAR10, as also done in the paper “Rethinking the value of network pruning” [1]. SFP pruned all layers in ResNet56 with a pre-defined uniform pruning rate across layers. For a fair comparison, we trained the masked ResNet56 by masking all the layers in each basic block (residual connection). Results show that our method and SFP (without fine-tuning) achieves 92.9% and 93.1% accuracy respectively, when both methods prune 30% of the weight parameters. (Note that our results on ResNet56 are a bit different from the numbers shown in Table 1 in our submission. For a fair comparison with L1-Pruning [6], we trained ResNet56 without masking the last Conv layer in each basic block, as what L1-Pruning did in the experiments.)
> We also show the sparsity patterns of the pruned networks for VGG19 (pruned 90% of the weight parameters, with accuracy 93.5% on CIFAR10) and ResNet56 (pruned 30% of the parameters, with accuracy 92.9% on CIFAR10). One can see that the pattern emerged from our method is similar to the pattern observed in the paper “Rethinking the value of network pruning”, e.g. for VGG19, late layers tend to have more redundancy than early layers, and for ResNet56, the pruning ratio tends to be uniform across stages.
>
> ============================================================= ========
> Layer-wise sparsity pattern for VGG19 with 90% of weight parameters pruned.
> Layer#	Filters ratio to be kept
> Conv 1	0.734
> Conv 2	0.984
> Conv 3	0.961
> Conv 4	0.992
> Conv 5	0.914
> Conv 6	0.758
> Conv 7	0.617
> Conv 8	0.520
> Conv 9	0.250
> Conv 10	0.172
> Conv 11	0.127
> Conv 12	0.115
> Conv 13	0.146
> Conv 14	0.145
> Conv 15	0.160
> Conv 16	0.361
> ============================================================= ========
> ============================================================= ========
> Stage-wise sparsity pattern for ResNet56 with 30% of weight parameters pruned.
> Layer#	Params ratio to be kept
> Stage 1	0.783
> Stage 2	0.681
> Stage 3	0.878
> ============================================================= ========
>
>
> Q3: What is the difference of your approach to [2]? They seem to be very similar. I think it is necessary to add some discussion in the related work. Is there any experimental results for comparison with [2]?
>
> Thanks for pointing us the AutoPruner paper, we will add it to our reference. Basically, AutoPruner proposed to prune weight by directly masking activation responses, rather than weight filters. Since activation responses are not learnable parameters, they add a AutoPruner layer to the network architecture. The AutoPrunner layer takes activation responses as input and the output is an approximate binary code in which zero value indicates the corresponding filter will be pruned. The AutoPrunner layer composes of average pooling,  max pooling, a fully-connected layer followed by a scaled sigmoid function, which introduces a large number of additional trainable parameters and computational complexity to the network.
> We would like to run AutoPruner experiments for comparisons, unfortunately, we didn’t find any source codes available on Github or the author’s website. We will probably implement the method by ourselves and add comparisons in the near future.

---

> > ### Author Response · Authors · 2019-11-13
> > **part 2 ==References==**
> >
> >
> > [1] Rethinking the Value of Network Pruning. https://arxiv.org/abs/1810.05270
> > [2] AutoPruner: An End-to-End Trainable Filter Pruning Method for Efficient Deep Model Inference. https://arxiv.org/abs/1805.08941
> > [3] Learning Efficient Convolutional Networks through Network Slimming https://arxiv.org/abs/1708.06519
> > [4] MorphNet: Fast & Simple Resource-Constrained Structure Learning of Deep Networks https://arxiv.org/abs/1711.06798
> > [5] Soft Filter Pruning for Accelerating Deep Convolutional Neural Networks. https://arxiv.org/abs/1808.06866
> > [6] Pruning Filters for Efficient ConvNets. https://arxiv.org/abs/1608.08710

---

### Official Review · AnonReviewer1 · 2019-10-24
**Official Blind Review #1**

**Rating:** 3

**Review:**

This paper proposes a masking process to improve the pruning of DNN. In addition, the algorithm proposes to automatically allocate the pruning rates over layers.

On the positive side:
- I do believe the main contribution is automatically allocating the pruning rates over the network.





Related works:
- How does this relate to methods using gating to prune in the presence of residual layers or BN?

- Paper claims multiple times that related works need a train-prune-retrain process which is only valid for post-processing works.

- I missed sparsity promoting works related to training from scratch where similar masks are implicitly used during training (even not explicitly stated) to maintain the zeros in the network. At least in [1,2] the training time is the same as the time used for training a network from scratch (same claim as in this paper).


Method:
- The mask is trained using a sigmoid function and claims this will be representative of the relevance of the 'neuron' within the layer. How is this really related?
- Why the initialization of the mask is to the mean? I think I am confused there. For initialization, I would argue all the neurons/parameters are relevant, right?

- The claim that this method is 'much simpler' is a bit subjective. I do not see why. Please elaborate.
- I am not sure if the claim of pruning filters and not considering the bn layer is correct. I would tend to think that, if pruning a neuron, the corresponding BN module should be modified (that is, propagating the zero to subsequent layers).

- I do like the extension to residual connections. It would be great to have more and clearer details on how is this done. Would this also apply to the UNET type of architecture?

- What is the intuition behind Eq 4 and how it is related to the relevance of a parameter?

- The extension to multiple metrics is of interest, however, there is little detailed there. How is the automatic allocation done? This is repeated in section 3.3 but details are missing.


- My understanding is that the regularization multiplier is affected by the learning rate, therefore their effect is lower as the training progresses. In this case, seems the opposite, right? (page 6 before 3.4).


- The part with the sparsity budget is interesting. What guarantees are that the newly enabled neurons are actually useful? Could it be possible that the budget suggested is not the right for the task at hand and, therefore, the additional parameters are not really relevant / needed?





Experiments:

- There is loss with little sparsity when it comes to Imagenet (15 and 17% pruning) does not seem very promising even the training time is similar.

- Seems like the experiments and comparisons to L1-pruning are not very surprising. It would be nice to have more comprehensive numbers. For imagenet, if using Resnet-50, would be easier to compare to other numbers.
- In the imagenet comparison, L1-pruning is the version-A of the paper. What about the others or why that particular model?
- How are the actual groups made?

Minor details:
- Please improve figure 1. It is not easy to understand. Same with Figure 3. What is seen in Figure 4.
- I do believe the WideResNet-28-10 number of parameters for BAR is not correct.
- Section 4.2 is a bit overselling. I do not see 'much-higher' parameter sparsity. The claims are mostly valid for VGG type of networks in this particular setting.




[1] Learning the Number of Neurons in Deep Networks, NeurIps 2016
[2] Compression-aware training of DNN, NeurIps 2017


**Experience Assessment:**

I have published in this field for several years.

**Review Assessment: Checking Correctness Of Derivations And Theory:**

I assessed the sensibility of the derivations and theory.

**Review Assessment: Checking Correctness Of Experiments:**

I carefully checked the experiments.

**Review Assessment: Thoroughness In Paper Reading:**

I read the paper thoroughly.

---

> ### Author Response · Authors · 2019-11-12
> **Author response to Reviewer #1**
>
> Thank you for reviewing our work. We answer your questions below:
>
> Q1: How does this relate to methods using gating to prune in the presence of residual layers or BN?
> Previous works prune weights/filters/layers via (1) L0/L1 regularization or variational dropout on weights/BN, or (2) simple binary thresholding defined over weight magnitudes / filters / BN norms. All these works can be treated as “hard” pruning through a gating function, where a hand-crafted threshold is required to make decision whether or not to prune. Thus, (iterative) fine-tuning is introduced to amortize the accuracy loss due to hard pruning.
> Our work differs from the prior art in that we perform “soft” pruning where a soft filter-wise mask with continuous values in [0,1] is automatically learned together with filter weight parameters, and filters are reparametrized by element-wise multiplications between filter mask and filter weights. Filters with mask value 0 are pruned away from the network architecture. Thus, fine-tuning after training is not required anymore, and also threshold is not required thanks to the use of Hard Sigmoid as the activation function for the mask layer.
>
>
> Q2: Paper claims multiple times that related works need a train-prune-retrain process which is only valid for post-processing works.
> Agree. We will update our draft to make the point more clearly.
>
>
> Q3: I missed sparsity promoting works related to training from scratch where similar masks are implicitly used during training (even not explicitly stated) to maintain the zeros in the network. At least in [1,2] the training time is the same as the time used for training a network from scratch (same claim as in this paper).
> Thanks for pointing us these 2 papers, we will add them to the reference. As mentioned in our draft, sparsity learning based pruning via L0/L1 regularization or variational dropout is similar to our work in the sense that they learn mask to prune the network. However, these works suffer from the problem that the sparsity is enforced for parameter size only, and not directly related to the target metric (e.g. FLOPs, or energy). While the mask defined in our work supports metrics beyond parameter size seamlessly.
>
>
> Q4: The mask is trained using a sigmoid function and claims this will be representative of the relevance of the 'neuron' within the layer. How is this really related?
> In early training all mask parameters are initialized as 0 and fed into hard-sigmoid activation function, the output mask is 0.5 for all filters, meaning that all filters are with equal importance. Like filter weight parameters, these mask parameters will also be continuously updated with SGD optimizer during training, until the model converges and meets the sparsity budget. According to the hard-sigmoid function, the final output mask is in the range of [0,1], important neurons will have mask output of 1, and unimportant neurons will have mask output of 0.
>
>
> Q5: Why the initialization of the mask is to the mean? I think I am confused there. For initialization, I would argue all the neurons/parameters are relevant, right?
> To clarify, the mask after Hard Sigmoid is all initialized as 0.5, indicating that all filters are with the equal importance; we think this is reasonable as there is no prior knowledge on which filters are more important than the others at the beginning of training.
> We also investigated other initialization methods for the mask, such as all mask initialized as 1 or random Gaussian. Results are comparable to initialization with 0.5. Another reason we initialize the mask with constant is we don't want to disrupt He/Xavier initialization.
>
>
> Q6: The claim that this method is 'much simpler' is a bit subjective. I do not see why. Please elaborate.
> We admit it is a bit subjective and will remove it accordingly. The message we would like to deliver is that the proposed mask module is light-weight and the additional computational cost introduced during training can be ignored. Moreover, by using shared BN mask, we did not need to retrain or fine-tune the pruned model, unlike previous work such as MorphNet[1] and Network Slimming[2]

---

> > ### Author Response · Authors · 2019-11-12
> > **Part 2**
> >
> >
> > Q7: I am not sure if the claim of pruning filters and not considering the bn layer is correct. I would tend to think that, if pruning a neuron, the corresponding BN module should be modified (that is, propagating the zero to subsequent layers).
> > Apologies for the ambiguity caused. We would like to clarify that the mismatch problem between Conv and BN layers exists for any filter pruning approaches that with threshold-based "hard" pruning. Given a Conv layer of a trained model, in the forward pass, filters with norm smaller than a pre-defined threshold are pruned away, the corresponding output activation maps with non-zero values are forced to be zero and fed to the successive BN layer. Similarly, in the backward pass, the gradients of output activation maps of this BN layer are also affected by its successive Conv layer with filters being pruned. Thus, the BN running mean/variance and trainable parameters (scale and shift) need to be adapted, due to the change of both the input and output activation maps of this BN layer, given that the BN statistics and parameters depend on all the input and output values respectively. This is the main reason that (iterative) fine-tuning is required after each "hard" pruning.
> >
> >
> > Q8: I do like the extension to residual connections. It would be great to have more and clearer details on how is this done. Would this also apply to the UNET type of architecture?
> > The main difficulty for pruning residual connections is the input activation maps for layers pointed to the element-wise addition layer must be aligned after filter pruning. There are 3 options for handling residual connections. First, as done in L1-pruning [4], one can skip the pruning of the shortcut and the last Conv layer in each basic block. In other words, there are no mask parameters to be learned for these layers. Second, the mask can be enforced to be shared between the layers pointed to the element-wise addition layer. Task ResNet56 v1 as an example, 2 mask layers are designed for a basic block with a shortcut path and another path composed of 2 consecutive Conv layers. One mask is for the first Conv layer, and the other mask is shared by the shortcut and the last Conv in the block. Third, one can also take a step forward to not share the mask between the shortcut and the last Conv in each basic block. Instead, we let the algorithm automatically search the optimal mask for each layer and store the mask in the pruned model, the mask vector is with dimension equals the number of output depth, in which 0 indicates the corresponding filter is pruned. When executing the element-wise addition operations, we align (i.e. reshape) the output activation maps using the mask for each layer independently, then sum the well-aligned activation maps together.
> >
> > In principle, mask sharing can be applied to UNET architecture as well. In this case a layer pair in U-residual connection must share the same mask to enforce the same output depth for residual addition operation to work.
> >
> >
> >  Q9: What is the intuition behind Eq 4 and how it is related to the relevance of a parameter?
> > Minimizing mean is essentially minimizing L1-norm on mask vector, and maximizing variance is soft-binarization [3] which pushes mask values to either 0 or 1. In this way, important neurons will quickly gain mask value of 1. Empirically we found that normalizing variance by dividing with mean is more reliable than variance alone. Otherwise the masks will go to 0 and 1 too quickly and somehow cannot be reversed.
> >
> >
> > Q10: The extension to multiple metrics is of interest, however, there is little detailed there. How is the automatic allocation done? This is repeated in section 3.3 but details are missing.
> > A simple heuristic we proposed is to average the mean and variance-to-mean of the masks from different metrics. For example, to optimize 2 metrics (parameter size and FLOP) simultaneously, we first compute the mean and variance-to-mean of the shared mask, which are multiplied by the dynamic \lambda parameters for each metric separately, then sum them together.
> > Algorithm 1 describes the main training steps for budget-aware MaskConvNet, in which \delta_s is approaching to 0 once the budget is met. The corresponding mask parameters are the automatic allocation solution for filter pruning. Section 3.4 explains how the algorithm re-activates pruned filters when the pruning rate exceeds the budget.

---

> > > ### Author Response · Authors · 2019-11-12
> > > **Part 3**
> > >
> > > Q11: My understanding is that the regularization multiplier is affected by the learning rate, therefore their effect is lower as the training progresses. In this case, seems the opposite, right? (page 6 ‪before 3.4‬). ‬‬
> > > Yes, the regularization multiplier is indirectly affected by the learning rate. As shown in Algorithm 1, the regularization multiplier is updated by \delta_s (\delta_s = target_sparsity – current_sparsity), and \delta_s is mainly driven by the mask values (output by Hard Sigmoid). The larger learning rate will drive the mask values to 0 faster. In our experiments, the learning rate is scheduled to be divided by 10 in certain epochs. Thus, the mask becomes sparse quickly in early epochs, and stabilizes later. Correspondingly, \delta_s moves to 0 quickly in early epochs, and so does the regularization multiplier.
> > >
> > >
> > > Q12: The part with the sparsity budget is interesting. What guarantees are that the newly enabled neurons are actually useful? Could it be possible that the budget suggested is not the right for the task at hand and, therefore, the additional parameters are not really relevant / needed?
> > > Our method doesn’t set any hand-crafted pruning criteria. Instead, with a pre-defined budget, the trainable mask layer enforces the back-propagation algorithm to automatically decide which filter is important and which is not, by updating the mask parameters. We observed that the pruned filters also follow the smaller-norm-less-informative rule, which is a common pruning criteria adopted by many prior works like Network Slimming.
> > > For any tasks, there is a trade-off between accuracy and pruning rate. In principle our algorithm is able to find the pruned networks that can meet any budget, via balancing the cross-entropy loss and regularization loss. The question is how to maintain the accuracy even for very high pruning rate, which depends on the redundancy of the original network architecture, the difficulty of the task as well as the data scale, etc.
> > >
> > >
> > > Q13: There is loss with little sparsity when it comes to Imagenet (15 and 17% pruning) does not seem very promising even the training time is similar.
> > > For a fair comparison with the L1-pruning [4] with ResNet34 on ImageNet, we didn’t prune the last Conv layer in each basic block (as done in L1-pruning). This is one reason the pruning rate is limited. Another reason is we found that for ImageNet task shallower ResNet is harder to prune as it is less redundant.
> > >
> > >
> > > Q14: Seems like the experiments and comparisons to L1-pruning are not very surprising. It would be nice to have more comprehensive numbers. For imagenet, if using Resnet-50, would be easier to compare to other numbers.
> > > Due to limited computation resources, we are still working on training on ImageNet with Resnet-50. Currently, we found that shallower ResNets are quite parameter-efficient, and putting high sparsity budget will severely degrade accuracy.
> > >
> > >
> > > Q15: In the imagenet comparison, L1-pruning is the version-A of the paper. What about the others or why that particular model?
> > > We could try all sparsity budgets but did not do, as training a ResNet model from scratch on ImageNet really takes a long time given that we only have limited computation resources. But we will definitely include more results once we have them.
> > >
> > >
> > > Q16: How are the actual groups made?
> > > Do you mean the groups in Table 1? We grouped by network architectures, i.e. VGG-16, VGG-19, ResNet56, ResNet-164, and WideResNet-28-10 as used in respective prior work on CIFAR10.
> > >
> > >
> > > Q17: I do believe the WideResNet-28-10 number of parameters for BAR is not correct.
> > > Thanks. We read the supplementary materials of the BAR paper and derived the numbers from there. We checked again and cannot find why the number of parameters is not correct. Would appreciate it if you can share more details and we will look into this accordingly.
> > >
> > >
> > > [1] Learning Efficient Convolutional Networks through Network Slimming https://arxiv.org/abs/1708.06519
> > > [2] MorphNet: Fast & Simple Resource-Constrained Structure Learning of Deep Networks https://arxiv.org/abs/1711.06798
> > > [3] PREDICTABILITY MINIMIZATION http://people.idsia.ch/~juergen/edgedetect/node2.html
> > > [4] Pruning Filters for Efficient ConvNets. https://arxiv.org/abs/1608.08710

---

### Public Comment · ~Anthony_Wittmer1 · 2019-10-28
**Formatting issue about the citations**

Hi, some citations are lacking brackets (\cite instead of \citep).

---

> ### Author Response · Authors · 2019-11-13
> **Thanks for the careful review**
>
> We will revise the citation format accordingly.

---

### Decision · Program_Chairs · 2019-12-19

**Decision:**

Reject

**Comment:**

This paper presents a method to learn a pruned convolutional network during conventional training.  Pruning the network has advantages (in deployment) of reducing the final model size and reducing the required FLOPS for compute.  The method adds a pruning mask on each layer with an additional sparsity loss on the mask variables. The method avoids the cost of a train-prune-retrain optimization process that has been used in several earlier papers.  The method is evaluated on CIFAR-10 and ImageNet with three standard convolutional network architectures.  The results show comparable performance to the original networks with the learned sparse networks.

The reviewers made many substantial comments on the paper and most of these were addressed in the author response and subsequent discussion.  For example, Reviewer1 mentioned two other papers that promote sparsity implicitly during training (Q3), and the authors acknowledged the omission and described how those methods had less flexibility on a target metric (FLOPS) that is not parameter size.  Many of the author responses described changes to an updated paper that would clarify the claims and results (R1: Q2-7, R2:Q3).

However, the reviewers raised many concerns for the original paper and they did not see an updated paper that contains the proposed revisions.  Given the numerous concerns with the original submission, the reviewers wanted to see the revised paper to assess whether their concerns had been addressed adequately. Additionally, the paper does not have a comparison experiment with state-of the art results, and the current results were not sufficiently convincing for the reviewers.  Reviewer1 and author response to questions 13--15 suggest that the experimental results with ResNet-34 are inadequate to show the benefits of the approach, but results for the proposed method with the larger ResNet-50 (which could show benefits) are not yet ready.

The current paper is not ready for publication.